# Efficacy of Catheter Ablation for Atrial Arrhythmias in Patients with Arrhythmogenic Right Ventricular Cardiomyopathy—A Multicenter Study

**DOI:** 10.3390/jcm10214962

**Published:** 2021-10-26

**Authors:** Alessio Gasperetti, Cynthia A. James, Liang Chen, Niklas Schenker, Michela Casella, Shinwan Kany, Shibu Mathew, Paolo Compagnucci, Andreas Müssigbrodt, Henrik K. Jensen, Anneli Svensson, Sarah Costa, Giovanni B. Forleo, Pyotr G. Platonov, Claudio Tondo, Jiang-Ping Song, Antonio Dello Russo, Frank Ruschitzka, Corinna Brunckhorst, Hugh Calkins, Firat Duru, Ardan M. Saguner

**Affiliations:** 1Department of Cardiology, University Heart Center, University Hospital Zurich, Rämistrasse 100, 8091 Zurich, Switzerland; sarah.costa@usz.ch (S.C.); frank.ruschitzka@usz.ch (F.R.); Corinna.Brunckhorst@usz.ch (C.B.); firat.duru@usz.ch (F.D.); ardan.saguner@usz.ch (A.M.S.); 2Cardiology and Arrhythmology Clinic, University Hospital “Ospedali Riuniti Umberto I—Lancisi—Salesi”, 60126 Ancona, Italy; michelacasella@hotmail.com (M.C.); paolocompagnucci1@gmail.com (P.C.); antonio.dellorusso@gmail.com (A.D.R.); 3Department of Biomedical Science and Public Health, Marche Polytechnic University, 60126 Ancona, Italy; 4Division of Cardiology, Department of Medicine, Johns Hopkins Hospital, Baltimore, MD 21205, USA; cjames7@jhmi.edu (C.A.J.); hcalkins@jhmi.edu (H.C.); 5State Key Laboratory of Cardiovascular Disease, Fuwai Hospital, Peking Union Medical College, Beijing 100000, China; chenliang@fuwaihospital.org (L.C.); fwsongjiangping@126.com (J.-P.S.); 6Department of Cardiology, Asklepios Klinik St. Georg Hamburg, 20099 Hamburg, Germany; nik.schenker@asklepios.com (N.S.); dr.mathew.shibu@googlemail.com (S.M.); 7Department of Clinical, Special and Dental Sciences, Marche Polytechnic University, 60126 Ancona, Italy; 8Department of Cardiology, University Heart and Vascular Centre Hamburg, University Medical Centre Hamburg-Eppendorf, 20251 Hamburg, Germany; s.kany@uke.de; 9Department of Electrophysiology, Heart Center University of Leipzig, 04289 Leipzig, Germany; andreas.muessigbrodt@gmail.com; 10Department of Cardiology, University Hospital of Martinique, 97200 Fort de France, Martinique, France; 11Department of Cardiology, Aarhus University Hospital, 8200 Aarhus, Denmark; hkjensen@clin.au.dk; 12Department of Clinical Medicine, Health, Aarhus University, 8200 Aarhus, Denmark; 13Department of Cardiology, Linköping University, 58183 Linköping, Sweden; anneli.svensson@gmail.com; 14Department of Medical and Health Sciences, Linköping University, 58183 Linköping, Sweden; 15Dipartimento di Cardiologia, ASST FBF-Sacco, 20149 Milano, Italy; forleo@me.com; 16Lund University Arrhythmia Clinic, Department of Cardiology, Skåne University Hospital, 22185 Lund, Sweden; pyotr.platonov@med.lu.se; 17Heart Rhythm Center, Centro Cardiologico Monzino, IRCCS, 20138 Milan, Italy; claudio.tondo@ccfm.it; 18Department of Clinical Sciences and Community Health, University of Milan, 20122 Milan, Italy

**Keywords:** arrhythmogenic right ventricular cardiomyopathy, atrial fibrillation, atrial flutter, pulmonary vein isolation, ablation in special populations

## Abstract

**Background:** Atrial arrhythmias are present in up to 20% of patients with arrhythmogenic right ventricular cardiomyopathy (ARVC). Catheter ablation (CA) is an effective treatment for atrial arrhythmias in the general population. Data regarding CA for atrial arrhythmias in ARVC are scarce. **Objective**: To assess the safety and efficacy of CA for atrial arrhythmias in patients with ARVC. **Methods:** In this international collaborative effort, all patients with a definite diagnosis of ARVC undergoing CA for atrial fibrillation (AF), focal atrial tachycardia (AT), or cavotricuspid isthmus (CTI)-dependent atrial flutter (AFl) were extracted from twelve ARVC registries. Demographic, periprocedural, and long-term arrhythmic outcome data were collected. **Results:** Thirty-seven patients were enrolled in the study (age 50.2 ± 16.6 years, male 84%, CHA_2_DS_2_VASc 1 (1,2), HAS-BLED 0 (0–2)). The arrhythmia leading to CA was AF in 23 (62%), focal left AT in 5 (14%), and CTI-dependent AFl in 9 (24%). Acute procedural success was achieved in all procedures but one (*n* = 1 focal left AT; 97% acute success). The median follow-up period was 27 (13–67) months, and 96%, 74%, and 61% of patients undergoing AF ablation were free from any atrial arrhythmia recurrence after a single procedure at 6 months, 12 months, and last follow-up, respectively. After focal AT ablation, freedom from atrial arrhythmia recurrence was 80%, 80%, and 60% at 6 months, 12 months, and last follow-up, respectively. All patients undergoing CTI ablation were free from atrial arrhythmia recurrences at 6 months, with 89% single-procedural arrhythmic freedom at last follow-up. One major complication (2.7%; PV stenosis requiring PV stenting) occurred. **Conclusions:** CA is safe and effective in managing atrial arrhythmias in patients with ARVC, with success rates comparable to the general population.

## 1. Introduction

Arrhythmogenic right ventricular cardiomyopathy (ARVC) is an inherited cardiomyopathy characterized by a progressive myocardial fibro-fatty infiltration (FFI), predominantly originating in the right ventricle (RV) [1]. Although ventricular dysfunction and ventricular arrhythmias represent the hallmarks of ARVC, a growing body of evidence from both ARVC animal models and patient cohorts has indicated atrial involvement [2,3,4]. The atria of patients with ARVC have been shown to be dilated and their function impaired, with the right atrium (RA) being predominantly affected by this process [3,4,5]. Considering electrocardiographic features, Platonov PG et al. first described an altered electrical conduction within the atria of patients with ARVC [6], and subsequent cohort studies reported a high rate of atrial arrhythmias in ARVC [3,7,8]. Particularly, atrial fibrillation (AF) and atrial flutter (AFl) were shown to be more frequent and presented at a younger age in patients with ARVC as compared to the general population [7,9].

Catheter ablation (CA) is an effective therapy for the treatment of atrial arrhythmias in the general population [10,11]. Data regarding the use, safety, and efficacy of CA in the management of atrial arrhythmias in ARVC are scarce, and previous work is limited to a few small sample-sized reports [12,13]. Therefore, the aim of this international multicenter study was to analyze the efficacy and safety of CA for the treatment of atrial arrhythmias in patients with ARVC.

## 2. Methods

The ARVC registries at 12 tertiary care institutions (University Hospital Zurich, Switzerland; Johns Hopkins Medical University, USA; IRCCS Centro Cardiologico Monzino, Italy; Asklepios Clinic St. George, Hamburg, Germany; Fuwai University Hospital, Beijing, China; Aarhus Hospital, Denmark; Skane University Hospital, Sweden; Linkoeping University Hospital, Sweden; Azienda Ospedaliera Luigi Sacco, Italy; University Clinic Hamburg-Eppendorf, Germany; University Hospital of Ancona, Italy; Heart Center University of Leipzig, Germany) were retrospectively searched for all consecutive ARVC patients fulfilling the inclusion criteria of the study. The earliest patient enrolled had his atrial procedure performed in 2014, while the last patient enrolled had her procedure performed in 2020.

The study was approved by the ethical boards of each contributing center, and informed consent was retrieved at each center in accordance with local and national regulations. This study was performed in accordance with the Declaration of Helsinki. The data that support the findings of this study are available from the corresponding author upon reasonable request.

### 2.1. Study Inclusion Criteria

Patients were enrolled in the present study upon the following inclusion criteria:-An ARVC diagnosis in accordance with the 2010 International Task Force Criteria (ITFC) [14]. Only patients reaching a “definite” diagnosis were included in the final cohort.-Performance of CA for an atrial arrhythmia of interest at any point during the clinical course; atrial arrhythmias of interest were AF, left-sided atrial tachycardia (AT), and CTI-dependent AFl. No ablation strategy or ablation energy source was pre-specified for study enrollment.-A minimum follow-up time of 6 months, with at least one follow-up visit with an arrhythmic assessment after CA.

No additional exclusion criteria were used.

Of note, the present study was aimed to assess the effectiveness of catheter ablation for left-sided atrial arrhythmias including AF in patients with ARVC. CTI-dependent AFl patients were also included due to it being the most common atrial arrhythmia in patients with ARVC and its common association with AF. Data of patients with right-sided atrial tachycardia other than CTI-dependent AFl, due to its rarity and relatively low clinical yield in this patient population, were not collected.

### 2.2. Data Collection

All available demographics, baseline, and pre-, peri- and post-procedural data (<7 days from the time of the procedure) were collected from patient medical records. Cardiac dimensions and function were retrieved, including data from echocardiography and cardiac magnetic resonance imaging, if available. If an adequate comparability in the obtained data was available, numerical values were reported. In the case of non-comparability, data were presented using a semiquantitative description (e.g., right/left atrium dimension: “normal”, “moderate dilation”, “severe dilation”). Imaging data were acquired according to current guidelines and/or local practice. Only membrane-active anti-arrhythmic drugs (Class I and Class III of the Vaughan Williams classification) were considered anti-arrhythmic drugs for the purpose of this study. Arrhythmic follow-up was assessed using 24 h, 48 h, and 7-day Holter ECGs; implantable loop recorders; or remote/in-person implantable cardioverter defibrillator (ICD) interrogation based on the clinical practice and expertise of each center. An ICD was available in 76% of the population for arrhythmic follow-up assessment and represented the most employed methodology. All data were pseudonymized and collected in a centralized database.

### 2.3. Study Definitions, Aims, and Outcomes

Acute success was defined as follows: (a) achievement of pulmonary vein isolation (PVI) in PVI-based procedures; (b) arrhythmia termination, if present at the time of CA, and/or non-inducibility of atrial arrhythmias in left-sided non-PVI-based procedures at the end of the procedure; (c) achievement of bidirectional block in CTI ablation procedures. Long-term success was defined as the absence of atrial arrhythmia recurrence during follow-up. An atrial arrhythmia recurrence at follow-up was defined as any atrial arrhythmia episode lasting >30 s after a 3-month post-procedural blanking period +/− antiarrhythmic medication.

The aim of this study was to assess the efficacy and safety of CA for the treatment of atrial arrhythmias in patients with ARVC. The primary outcome of the study was freedom from atrial arrhythmia recurrence at 6 months. Secondary outcomes included mid-term (12-month follow-up) and long-term (last available follow-up) outcome data as well as periprocedural complication rate and changes in pharmacological therapy in the long term.

### 2.4. Statistical Analysis

All statistical analyses were performed using STATA v. 14.0 (STATACorp, College Station, TX, USA).

Distribution of all continuous variables was tested for normality using the Shapiro–Wilk test. Normally distributed variables are reported as mean ± standard deviation, while non-normally distributed data are reported as median (with interquartile ranges (IQR)). Categorical variables are expressed as counts (percentage). Kaplan–Meier estimates were used to report arrhythmic outcome data.

## 3. Results

### 3.1. Patient Population

A total of 37 patients with definite ARVC (32 probands and 5 family members) fulfilling the inclusion criteria were enrolled in this study. ARVC diagnosis was reached at a mean age of 50.2 ± 16.6 years. ARVC diagnostic characteristics are shown in Table 1.

The atrial arrhythmia treated by CA was AF in 23 (62%) patients, left-sided focal AT in 5 (14%) patients, and CTI-dependent AFl in 9 (24%) patients, respectively. The first atrial arrhythmia episode was observed at a mean age of 48.7 ± 15.3 years, and in most cases, its presentation preceded the diagnosis of ARVC (median difference between atrial arrhythmia presentation and ARVC diagnosis: −2 (−4; 0) years). Baseline characteristics are summarized in Table 2.

### 3.2. Periprocedural Data

Patients underwent CA less than 1 year (0–2) after their first atrial arrhythmia episode (mean age at CA: 49.0 ± 15.4 years). Three patients (8%) were referred for CA after an inappropriate ICD discharge that was triggered by an atrial arrhythmia. Data for RA and left atrial (LA) dimensions were available for 30 (81%) and 32 (86%) patients, respectively. The majority of patients (29/32) had a normal LA size, while moderate or severe RA dilation was observed in 43% of cases.

### 3.3. Procedural Data

A total of 23 (62%) ARVC patients underwent PVI. In 16 (43%) of them, only circumferential PVI was performed. The other seven patients had PVI plus additional ablation targets: in four cases (11%), an anterior mitral isthmus line was performed; in two cases (5%), posterior wall isolation was performed; and in one case (3%), a roof line was additionally performed. In six patients undergoing PVI, CTI ablation was additionally performed. In 22 (96%) patients with AF, radiofrequency energy was used for PVI, while cryoballoon was used in 1 patient (4%). Five patients (14%) underwent left-sided atrial ablation (focal ablation in four and anterior mitral isthmus line in one) without PVI, while in the remaining nine patients (24%), CTI ablation was performed.

### 3.4. Acute Procedural Data

Acute procedural success was achieved in 36 of 37 patients (97%). Catheter ablation failed in one patient with a focal left-sided AT. Six procedures required an intra-operative electrical cardioversion. Two periprocedural complications (5.4%) were encountered: a left inferior PV stenosis (after PVI) requiring subsequent stenting in one patient was considered a major complication, whereas a groin hematoma after AFl ablation in another patient, which was managed conservatively without further sequelae, was considered a minor complication. Periprocedural and procedural data are reported in Table 3.

### 3.5. Long-Term Outcome Data

All patients were followed up for a minimum of 6 months. The median follow-up period of the cohort was 27 (13–67) months.

Figure 1 displays single-procedural freedom from atrial arrhythmia recurrence rates +/− AADs, stratified by procedure type at 12 months. At 6-months follow-up, 96% of patients with AF undergoing PVI-based procedures were free from arrhythmic recurrences, while at 12 months, their freedom from atrial arrhythmias was 74%. At last available follow-up, 25/37 patients (68%) were free from arrhythmic recurrences, with 2 patients being on a previously failed AAD (*n* = 2 Class III). Of patients undergoing left-sided non-PVI-based procedures, 80% (four out of five) were free from recurrences at both the 6- and 12-month follow-up after a single procedure. At last available follow-up, 60% (three out of five) of the patients were free from atrial arrhythmias (*n* = 1 patient on a class Ic AAD). Among the nine patients who underwent ablation of CTI-dependent AFl, 100% (nine out of nine) were free of recurrences at 6 months and 89% (eight out of nine) at 12 months and last follow-up after a single procedure (*n* = 2 patients on a class III AAD).

During follow-up, a total of seven patients (19%) underwent a repeat procedure (*n* = 1 for CTI-dependent AFl, *n* = 1 for left-sided focal AT, and *n* = 5 for AF). When including repeat procedures, all patients with CTI-dependent AFl were free from any atrial arrhythmia at last follow-up. At last follow-up, 24% and 46% of the population were on anti-arrhythmic drugs and on oral anticoagulation, respectively (no significant changes from baseline, *p* = 0.56 and *p* = 0.12, respectively). No severe bleeding events or cerebral vascular events were observed. The overall outcome data are summarized in Table 4.

## 4. Discussion

This study represents the largest systematic assessment of the efficacy and safety of CA for the treatment of atrial arrhythmias inpatients with ARVC to date.

The main findings of this international multicenter study are: (a) CA was an effective strategy for the management of atrial arrhythmias in patients with ARVC, with high acute procedural success and low complication rates; (b) CA achieved considerable long-term freedom from atrial arrhythmia recurrences, both for PVI and non-PVI-based ablative procedures; (c) at the time of CA, the majority of patients with ARVC and atrial arrhythmias presented with normal LA dimensions, whereas RA dimensions were increased in more than one-third.

### 4.1. Atrial Arrhythmias and ARVC

Up to 15–20% of ARVC patients enrolled in large and well-characterized cohorts were reported to suffer from episodes of atrial arrhythmias, with AF being the most common one [3,7,8]. Atrial arrhythmias in these patients may develop as part of a primary pathogenic process (atrial cardiomyopathy) due to desmosomal/connexome impairment in the atria or as a consequence of atrial dilation following depressed ventricular function and dilation.

The presence of atrial arrhythmias in patients with ARVC has important clinical consequences, such as an increased risk of inappropriate ICD discharges [3,15]. Of the 35 ARVC patients with atrial arrhythmias and an ICD reported by Chu et al. [12], 3 presented with inappropriate ICD therapies triggered by an atrial arrhythmia episode. A similar rate (8%) of atrial arrhythmia-triggered inappropriate shocks was also found in our cohort, representing an important indication for the referral for CA in these patients. Additionally, the presence of atrial arrhythmias (AF in particular) has been strongly associated with an increase in life-threatening ventricular arrhythmic events in a study from Mazzanti et al. [16]. Although most of the attention in this specific subset of patients has been paid to ventricular arrhythmias, an adequate clinical management of patients with ARVC undoubtedly also requires an appropriate management of atrial arrhythmias.

### 4.2. Outcomes of Catheter Ablation of Atrial Arrhythmias in ARVC

CA is an effective and safe therapeutic option for the management of atrial arrhythmias in the general population [11]. To the best of our knowledge, prior evidence regarding the safety and efficacy of CA for atrial arrhythmias in patients with ARVC is limited to two small-scale studies. Chu et al. [12] reported feasibility data from 7 patients with ARVC suffering from CTI-dependent AFl undergoing CA for rhythm control, while Cardona-Guarache et al. reported safety and feasibility of AA ablation in 11 patients (*n* = 3 PVI; *n* = 2 atypical flutter; *n* = 5 typical flutter; *n* = 1 AT).

In our study, 23 ARVC patients underwent PVI and/or extended procedures for the management of AF/left-sided AT. Patients responded well to this intervention, achieving similar single-procedural rates of freedom from AF/left-sided AT as those reported in the general population. The rates of freedom from atrial arrhythmias recurrence at 6-month, 12-month, and at last available follow-up were 96%, 74%, and 68%, respectively. The predominance of paroxysmal AF, the normal LA size (in 91%) and preserved LVEF, and the relatively young age at the time of CA probably represented factors positively influencing outcome. Five patients underwent CA for a focal, left-sided AT, showing 60% single-procedural freedom from arrhythmias at last follow-up. Nine patients underwent CTI ablation for typical AFl. Although a dilated RA was present in 43% of the study population, CTI ablation was also very effective in this population, with eight out of nine patients being free from atrial arrhythmias after a single procedure at last follow-up. The ninth patient required repeat CTI ablation, after which no more atrial arrhythmias were observed. Complication rates except for PV stenosis (2.7%) were low and similar to those reported in the general population referred for CA of atrial arrhythmias [17]. The rather high rate of PV stenosis was probably related to the small sample size of the overall population.

Our data suggest that patients with ARVC should not be precluded from CA for atrial arrhythmias in light of their underlying cardiac disease and that the management of AF/AFl should be similar to that offered to the general population. These findings could additionally result in particular interest for two reasons.

First, since atrial arrhythmias are associated with inappropriate ICD discharges and adverse outcomes in ARVC patients, performing CA in this population may improve outcomes [18]. Of note, ARVC patients may undergo other invasive procedures (such as electrophysiological studies, voltage mapping-guided biopsies, or CA for ventricular arrhythmias), and CA for atrial arrhythmias may be potentially performed during the same procedure, reducing the need for multiple interventions and increasing the overall procedural benefit. Second, PVI currently represents the gold-standard lesion set for AF management. Despite the pathophysiology of AF being incompletely understood, PVI seems to be not only effective in the general population but also in channelopathies (e.g., Brugada syndrome [19]) and cardiomyopathies (e.g., hypertrophic cardiomyopathy [20]). The observation that PVI provided a good long-term outcome in our study suggests that AF in this population is driven by similar mechanisms as in the general population. Obtaining data regarding the efficacy of CA for atrial arrhythmias in specific subpopulations may provide important knowledge for refining our understanding of both atrial arrhythmias and of the underlying disease itself.

### 4.3. Limitations

There were several limitations to our study. First, given the rarity of patients with ARVC, particularly those presenting with symptomatic AA, the number of patients included in this study is small. Second, given the retrospective nature of the study, it was not possible to uniformly assess echocardiographic imaging criteria and pursue arrhythmic follow-up strategies due to differences in clinical practice of participating centers. Third, cardiac magnetic resonance imaging (CMR) with a specific protocol assessing atrial fibrosis was not available in our cohort. Finally, it should also be noted that all centers involved in the study were third-level referral centers, and, therefore, some degree of selection and referral bias cannot be ruled out. Further studies are needed to confirm generalizability of this data in an all-comer population of patients with ARVC.

## 5. Conclusions

CA represents an effective and safe therapeutic option for the management of atrial arrhythmias in patients with ARVC, with overall long-term success rates comparable to those reported in the general population. Larger prospective studies are needed to confirm these findings.

## Figures and Tables

**Figure 1 jcm-10-04962-f001:**
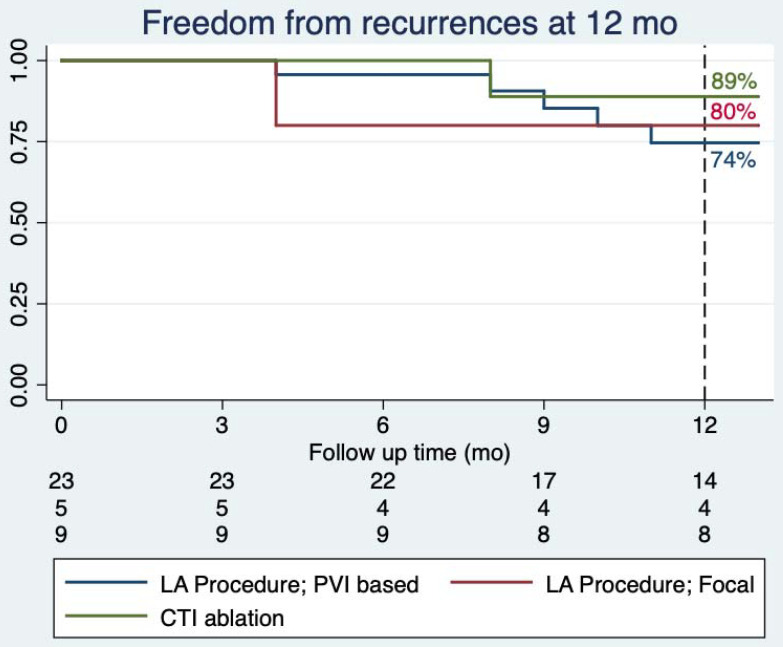
Single-procedural freedom from atrial arrhythmia recurrences stratified by ablation procedure performed at 12 months. CTI: cavotricuspid isthmus; LA: left atrial; PVI: pulmonary vein isolation.

**Table 1 jcm-10-04962-t001:** ARVC diagnostic characteristics of the study cohort.

ARVC Diagnostic Characteristics	
Age at ARVC diagnosis (years), mean ± s.d.	50.2 ± 16.6
Sustained VAs before/at ARVC diagnosis, *n* (%)	24 (65)
ARVC score, median [IQR]	5 (4–6)
Category I—Global or regional dysfunction	
Major, *n* (%)	20 (54)
Minor, *n* (%)	11 (30)
Category II—Tissue characterization of the wall	
Major, *n* (%)	4 (11)
Minor, *n* (%)	3 (8)
Category III—Repolarization abnormalities	
Major, *n* (%)	13 (35)
Minor, *n* (%)	9 (26)
Category IV—Depolarization abnormalities	
Major, *n* (%)	10 (27)
Minor, *n* (%)	12 (32)
Category V—Ventricular arrhythmias	
Major, *n* (%)	19 (51)
Minor, *n* (%)	16 (43)
Category VI—Family history	
Major, *n* (%)	3 (8)
Minor, *n* (%)	4 (11)
Genetic testing performed, *n* (%)	13 (35)
Negative, *n* (%)	6 (46) *
Positive, *n* (%)	7 (54) *
PKP-2, *n* (%)	5 (71) *
DSG-2, *n* (%)	2 (29) *

DSG-2: desmoglein-2; IQR: interquartile range; PKP-2: plakophilin-2; VA: ventricular arrhythmias. * Relative percentage calculated using as denominator the number of available data.

**Table 2 jcm-10-04962-t002:** Baseline clinical characteristics for the study cohort.

Baseline Characteristics (Overall Cohort, *n* = 37)	
Male, *n* (%)	31 (84)
Age at first atrial arrhythmia presentation (years), mean ± s.d.	48.7 ± 15.3
First atrial arrhythmia type	
Atrial fibrillation, *n* (%)	23 (62)
Paroxysmal, *n* (%)	16 (43)
Persistent, *n* (%)	5 (14)
Long-standing persistent, *n* (%)	2 (5)
Focal left-sided atrial tachycardia, *n* (%)	5 (14)
Cavotricuspid isthmus-dependent atrial flutter, *n* (%)	9 (24)
Probands, *n* (%)	32 (87)
Athletes, *n* (%)	7 (19)
BSA (m^2^), median [IQR]	1.9 (1.8–2.1)
CHA2DS2VASc, median [IQR]	1 (1,2)
Age > 64 y.o., *n* (%)	7 (19)
Age > 74 y.o., *n* (%)	2 (5)
History of CHF, *n* (%)	15 (41)
Female gender, *n* (%)	6 (16)
Hypertension, *n* (%)	12 (32)
Previous stroke/TIA, *n* (%)	3 (8)
Vascular disease history, *n* (%)	4 (11)
Diabetes, *n* (%)	3 (8)
HAS-BLED, median [IQR]	1 (0–2)
Age > 65 y.o, *n* (%)	9 (24)
Hypertension, *n* (%)	12 (32)
Renal disease, *n* (%)	4 (11)
Liver disease, *n* (%)	0
Stroke history, *n* (%)	3 (8)
Prior major bleeding, *n* (%)	1 (3)
Labile INR, *n* (%)	1 (3)
Medication predisposing bleeding, *n* (%)	14 (38)
Alcohol abuse, *n* (%)	3 (8)
EHRA score, median [IQR]	1 (0–2)
ICD, *n* (%)	28 (76)
Inappropriate discharge due to atrial arrhythmia, *n* (%)	3 (8)

BSA: body surface area; CHF: congestive heart failure; ICD: implantable cardioverter–defibrillator; INR: international normalized ratio.

**Table 3 jcm-10-04962-t003:** Periprocedural and procedural characteristics of the study cohort.

Periprocedural and Procedural Characteristics	
Age at CA (years), mean ± s.d.	49.0 ± 15.4
Pharmacological therapy at the time of CA	
Oral anticoagulation, *n* (%)	23 (62)
VKA, *n* (%)	10 (27)
DOAC, *n* (%)	13 (35)
Anti-arrhythmic therapy, *n* (%)	11 (30)
Class Ic, *n* (%)	3 (8)
Class III, *n* (%)	8 (22)
Morphology and functional data at the time of CA	
Right atrial dimension available, *n* (%)	30 (81)
Normal dimension, *n* (%)	17 (57) *
Moderate dilation, *n* (%)	9 (30) *
Severe dilation, *n* (%)	4 (13) *
Left atrial dimension available, *n* (%)	32 (86)
Normal dimension, *n* (%)	29 (91) *
Moderate dilation, *n* (%)	1 (3) *
Severe dilation, *n* (%)	2 (6) *
RVEF available, *n* (%)	21 (57)
Normal (>49%), *n* (%)	4 (19) *
Reduced (26–49%), *n* (%)	15 (71) *
Severely reduced (<26%), *n* (%)	2 (10) *
LVEF (%), mean ± s.d.	52.4 ± 6.8
Area targeted during CA	
Pulmonary vein isolation, *n* (%)	23 (62)
PVI only, *n* (%)	16 (43)
PVI + anterior mitral line, *n* (%)	4 (11)
PVI + posterior wall isolation, *n* (%)	2 (5)
PVI + roof line, *n* (%)	1 (3)
Non-PVI-based left-sided procedures, *n* (%)	5 (14)
Focal ablation, *n* (%)	4 (11)
Anterior mitral line, *n* (%)	1 (3)
Stand-alone CTI ablation, *n* (%)	9 (24)
Power settings for ablation, Watts [IQR]	40 (35–40)
Procedural time	
Left-sided procedure (min), median [IQR]	140 (105–143)
Right-sided procedure (min), median [IQR]	70 (60–110)
Fluoroscopy time	
Left-sided procedure (min), median [IQR]	18.5 (11.7–19.5)
Right-sided procedure (min), median [IQR]	13.5 (12.0–14.0)
Acute Success, *n* (%)	36 (97)
Pulmonary vein isolation, *n* (%)	23 (100)
Non PVI-based left-sided procedure, *n* (%)	4 (80)
Stand-alone CTI, *n* (%)	9 (100)
Need for ECV during procedure, *n* (%)	6 (16)
Complications, *n* (%)	2 (5.4)
Vascular access hematoma, *n* (%)	1 (2.7)
PV stenosis, *n* (%)	1 (2.7)

* Relative percentage calculated using the number of available data as denominator. CA: catheter ablation, CTI: cavotricuspid isthmus; DOAC: direct oral anticoagulants; ECV: electrical cardioversion; LA: left atrium; LVEF: left ventricular ejection fraction; PV: pulmonary vein; RA: right atrium; RVEF: right ventricular ejection fraction; VKA: vitamin K antagonists.

**Table 4 jcm-10-04962-t004:** Cohort outcomes at follow up.

Outcome Analysis	
Follow-up time (months), median [IQR]	27 (13–67)
Single-procedural freedom from any atrial arrhythmia	
At 6 months, *n* (%)	35/37 (95)
After PVI, *n* (%)	22/23 (96)
After non-PVI-based left-sided procedure, *n* (%)	4/5 (80)
After CTI ablation, *n* (%)	9/9 (100)
At 12 months, *n* (%)	26/33 (79)
After PVI, *n* (%)	14/19 (74)
After non-PVI-based left-sided procedure, *n* (%)	4/5 (80)
After CTI ablation, *n* (%)	8/9 (89)
At last available follow-up, *n* (%)	25/37 (68)
After PVI, *n* (%)	14/23 (61)
After non-PVI-based left-sided procedure, *n* (%)	3/5 (60)
After CTI ablation, *n* (%)	8/9 (89)
Pharmacological therapy at last available follow-up	
Oral anticoagulation, *n* (%)	17 (46)
VKA, *n* (%)	8 (22)
DOAC, *n* (%)	9 (24)
Anti-arrhythmic therapy, *n* (%)	9 (24)
Class Ib, *n* (%)	1 (3)
Class Ic, *n* (%)	5 (14)
Class III, *n* (%)	3 (8)
LVEF at last available follow-up, mean ± standard deviation	49.4 ± 12.4
Major bleeding during follow-up, *n* (%)	0
Stroke/TIA during follow-up, *n* (%)	0
Inappropriate ICD shock due to atrial arrhythmias at follow-up, *n* (%)	0
Patients undergoing a repeat procedure, *n* (%)	7 (19)
PVI, *n* (%)	3 (8)
Anterior mitral line, *n* (%)	2 (5)
Focal ablation, *n* (%)	1 (3)
CTI ablation, *n* (%)	1 (3)

## Data Availability

Data supporting the current study are available upon reasonable request to the corresponding author.

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
