# Peer review of "Efficacy of Catheter Ablation for Atrial Arrhythmias in Patients with Arrhythmogenic Right Ventricular Cardiomyopathy—A Multicenter Study"

_jcm, 2021, doi:10.3390/jcm10214962_

Round 1
Reviewer 1 Report
This is a small multicentric observational study of 37 patients with ARVC with various atrial arrhythmias who were treated with catheter ablation. The main finding, according to the authors of the study, was that catheter ablation appeared to be safe and effective in patients undergoing PVI, left- or right-sided flutter ablation. 19% of the patients (mainly PVI) underwent a repeat procedure during the follow-up period of (median) 27 months, and 24% of patients were under ungoing antiarrhythmic drug therapy (mainly class Ic AAD). These results are in line with published literature in other patient groups. In most patients catheter ablation was performed before diagnosis of ARVC, so the presented ARVC diagnostic characteristics likely may not fully reflect clinical status of the patients at the time of ablation.
Main limitations:
- small sample size and combination of various right- and left-sided procedures from multiple centers
- lack of a control group
Remarks and questions:
A PV stenosis rate of 2,7% would be unacceptably high for a contemporary cohort. However, due to the small sample size, the significance of calculating complication rates is limited.
Electroanatomic mapping was used to assess the amount of atrial fibrosis, however only few details are provided. Was atrial mapping performed before or after treatment? Was it performed in Sinus rhythm? As data on atrial mapping for fibrosis was available only in a minority of patients in this already small-sized and inhomogeneous sample, elimination of this aspect from the study might be warranted.
Figure 2 can be omitted as it is not directly related to the research findings.
Even when considering the shortcomings of the current study, the statement of the authors, that ARVC patients should not be precluded from catheter ablation, seems reasonable.
Reviewer 2 Report
In this multicenter, retrospective study, Gasperetti and collaborators describe ARVC patients who undergo left atrial arrhythmias or typical flutter ablation. Overall, 37 patients are collected from 12 centers: 23 had undergone AF ablation, left atrial tachycardia in 5, and typical flutter in 9. Atrial arrhythmias were diagnosed, on average, two years before a definite ARVC diagnosis. Acute success was reached in 36/37 patients, while long-term efficacy was high in the three groups (freedom from recurrent arrhythmia 80%-96%).
The manuscript is well written. Previous literature on catheter ablation for atrial arrhythmias in patients with ARVC is scarce, and the authors are to be commended for their efforts to recruit this cohort.
The main limitations of the present manuscript are the very low sample size and the very remarkable heterogeneity, including three types of arrhythmias and, possible, many operators from 12 centers.
I have some concerns of the methods. When were patients recruited? Between which years were they recruited?
The authors recognize that patients with ARVC show a more intense right atrial structural remodeling than left atrial remodeling (line 62). It is therefore unclear why patients with left atrial tachycardia were included, but right atrial arrhythmias were excluded (except for typical flutter).
The fact that on average 3 patients per each center were included, all in centers with a recognized expertise in ARVC, suggests that this is a very selected population. Extrapolation to all patients with ARVC is questionable.
The authors collect data on electroanatomical maps. Results are shown in Table 2 but are not discussed anywhere in the text. Moreover, more information is needed for this analysis. How many points were obtained per patients? Were they homogeneously distributed throughout the atria?
How were left and right atrial measurements categorized? This is an important issue, since data were obtained from many different centers and imaging tests may use different criteria.
Some of the variables reported in the text are of limited interest and are also reported in table, making the text repetitive more difficult to follow (e.g., the use of antiarrhythmics and anticoagulants in lines 162-165).
Author Response
REVIEWER #2
In this multicenter, retrospective study, Gasperetti and collaborators describe ARVC patients who undergo left atrial arrhythmias or typical flutter ablation. Overall, 37 patients are collected from 12 centers: 23 had undergone AF ablation, left atrial tachycardia in 5, and typical flutter in 9. Atrial arrhythmias were diagnosed, on average, two years before a definite ARVC diagnosis. Acute success was reached in 36/37 patients, while long-term efficacy was high in the three groups (freedom from recurrent arrhythmia 80%-96%).The manuscript is well written. Previous literature on catheter ablation for atrial arrhythmias in patients with ARVC is scarce, and the authors are to be commended for their efforts to recruit this cohort.
R: We thank Reviewer #2 for her/his keen comments and the Editorial Board members for giving us the chance of addressing them.
The main limitations of the present manuscript are the very low sample size and the very remarkable heterogeneity, including three types of arrhythmias and, possible, many operators from 12 centers. I have some concerns of the methods. When were patients recruited? Between which years were they recruited?
R: We thank the reviewer for highlighting this important aspect. Patients were extracted from long-term running major national ARVC registries. The earliest patient enrolled had his atrial procedure done in 2014, while the last patient enrolled had her procedure done in 2020. We have added this information in the methods section of the revised manuscript (page 5, first paragraph).
The authors recognize that patients with ARVC show a more intense right atrial structural remodeling than left atrial remodeling (line 62). It is therefore unclear why patients with left atrial tachycardia were included, but right atrial arrhythmias were excluded (except for typical flutter).
R: We thank the reviewer for highlighting this point. This manuscript was originally thought as an assessment of left sided atrial arrhythmia ablation, in particular pulmonary vein isolation for treating atrial fibrillation, but also left-sided atrial tachycardia/atypical flutter in patients with ARVC, due to the lack of data addressing this important topic in the literature. Among co-authors, we then decided to include CTI dependent flutter procedures as well, which is the most common atrial arrhythmia in ARVC, and hence due to the clinical importance of the arrhythmia and its common association with atrial fibrillation. Moreover, in our international Zurich ARVC registry, we have not yet encountered an ARVC patient with a right-sided atrial arrhythmia other than CTI-dependent flutter thus far.
The fact that on average 3 patients per each center were included, all in centers with a recognized expertise in ARVC, suggests that this is a very selected population. Extrapolation to all patients with ARVC is questionable.
R: We thank the reviewer for this important comment. We fully agree with the reviewer that this is a highly selected population of ARVC patients. Nevertheless, this is the worldwide largest assembled ARVC cohort thus far investigating the outcome of catheter ablation for atrial arrhythmias in ARVC. This point has been added to the limitation section, pointing out the need for further studies allowing generalizability of this data to all patients with ARVC.
The authors collect data on electroanatomical maps. Results are shown in Table 2 but are not discussed anywhere in the text. Moreover, more information is needed for this analysis. How many points were obtained per patients? Were they homogeneously distributed throughout the atria?
R: As Reviewer #2 mentions, EAM data present in the original draft was partial since this data was not available for many ablation procedures. Since our cohort is limited by its small sample size and as per Reviewer#1 request, EAM data have been removed from the manuscript, to improve manuscript clarity.
How were left and right atrial measurements categorized? This is an important issue, since data were obtained from many different centers and imaging tests may use different criteria.
R: We thank the reviewer for highlighting this important issue. We fully agree that due to the retrospective and multicenter nature of this work, there may have been some inhomogeneities in echocardiographic measurements. Moreover, as mentioned in the results section of the first draft, data for right atrial (RA) and left atrial (LA) dimensions were only available for 30 (81%) and 32 (86%) patients, respectively. Unfortunately, this is a common problem in multicenter studies investigating echocardiographic dimensions in ARVC including very important ones. In our study, the majority of patients (29/32) had a normal LA size, while moderate or severe RA dilation was observed in 43% of cases. Nevertheless, in all centers except the Beijing Fuwai center, LA and RA dimensions were assessed according to current echocardiographic guidelines by the Northern American/European associations. We have added this aspect in the methods and limitation sections of the revised manuscript.
Some of the variables reported in the text are of limited interest and are also reported in table, making the text repetitive more difficult to follow (e.g., the use of antiarrhythmics and anticoagulants in lines 162-165).
R: We thank the reviewer for this comment. The results section has been amended as per Reviewer #2 suggestions, with redundancy between text and associated tables being removed.
Please see attachment

Round 2
Reviewer 2 Report
The authors have replied most of my comments. I would only add that the authors reply and justification for only including left atrial arrhythmias and CTI ablation should be included in the main text.
Author Response
Dear Reviewer#2,
Thank you for your comment. As requested, we have implemented the justification for only including left atrial arrhythmias and CTI ablation in the body of the reviewed manuscript (see attached).
Best and thanks,
Alessio
